# Bridging self-directed learning and competency in interdisciplinary teaching among STEM lecturers: How psychological capital connects

**Wenhao Su**[☉], **Hidayah Mohd Fadzil**[ID][☉]*, **Rose Amnah Abd. Rauf**[☉]

Faculty of Education, University of Malaya, Kuala Lumpur, Malaysia

☉ These authors contributed equally to this work.
* hidayahfadzil@um.edu.my

## Abstract

University STEM education is vital for cultivating innovative talents but is hindered by a lack of lecturers skilled in interdisciplinary teaching. Research on the interplay between psychological capital (PsyCap), competency in interdisciplinary teaching (CIT), and self-directed learning (SDL) is limited. This study investigates a mediation model, exploring PsyCap's role between SDL and CIT among 427 Chinese STEM lecturers using proportional stratified random sampling and partial least squares structural equation modeling (PLS-SEM). The results revealed that SDL had a significant direct effect on CIT ($\beta = 0.210$, $t = 4.429$, $p < 0.001$) and PsyCap ($\beta = 0.603$, $t = 16.737$, $p < 0.001$). PsyCap also significantly influenced CIT ($\beta = 0.546$, $t = 12.748$, $p < 0.001$). Mediation analysis further indicated that PsyCap partially mediated the relationship between SDL and CIT, with a significant indirect effect ($\beta = 0.329$, $t = 9.738$, $p < 0.001$). These findings contribute to the literature by validating a novel mediation model integrating SDL and PsyCap to explain CIT. It also offers practical implications for offering practical pathways for improving the training of STEM lecturers, with SDL and PsyCap identified as key leverage points for enhancing the quality of interdisciplinary teaching in universities. Given the study's single-mediator design, future research may adopt moderated mediation models to explore when and how SDL and PsyCap more strongly influence interdisciplinary teaching competency.

## Introduction

STEM education lies at the heart of innovation, driving economic growth and development while addressing global challenges [1]. The proactive promotion and exploration of STEM education across various educational stages have become a key component of global talent development strategies [2]. In this developmental process, interdisciplinary teaching has emerged as a critical factor in advancing

**Data availability statement:** All data files are available from the FigShare database (DOI: https://doi.org/10.6084/m9.figshare.28830521).

**Funding:** The author(s) received no specific funding for this work.

**Competing interests:** The authors have declared that no competing interests exist.

STEM education [3]. Researches indicate that STEM teachers play a catalytic role in fostering students' interdisciplinary skills, with their competency in interdisciplinary teaching directly influencing students' learning outcomes and career development [4]. However, existing STEM teachers, under the high pressure of research performance evaluations, often develop a utilitarian mindset that prioritizes research over teaching [5,6]. This results in a lack of interdisciplinary teaching knowledge, making it difficult to effectively implement STEM education [7]. Furthermore, the homogeneous pre-service and in-service training systems for teachers fail to meet the actual interdisciplinary teaching needs of STEM educators, posing another significant barrier to building a robust STEM teaching workforce [8]. In contrast to developed countries like the United States, the United Kingdom, Germany, Finland, and Japan, which have established distinctive interdisciplinary talent cultivation systems, China's research on STEM teachers' competency in interdisciplinary teaching mainly focuses on the K-12 stage [9,10]. This approach neglects the study of factors, mechanisms, and strategies for enhancing the competency in interdisciplinary teaching of university STEM lecturers.

According to international standards and specific national or regional regulations, the student-to-lecturer ratio in higher education generally ranges from 10:1–20:1 [11]. In China, the requirement for the student-to-lecturer ratio in universities usually does not exceed 18:1 [12]. Currently, there are 47.63 million university students in China, with approximately 16 million studying STEM disciplines [13]. However, there are only about 308,000 STEM lecturers in the STEM field, resulting in a student-to-lecturer ratio as high as 52:1 [14]. This indicates a significant shortage of STEM lecturers, and those who possess the competency for interdisciplinary teaching are even more scarce. To specifically enhance STEM lecturers' competency in interdisciplinary teaching, the current research perspective has shifted from relying on planned and organized training activities within the workplace to focusing on the intrinsic motivation of lecturers in informal settings. Studies have shown that self-directed learning and teaching competency are symbiotic [15,16]. Lecturers with a high propensity for self-directed learning tend to have higher teaching efficacy and capability [17]. This is because they are better at formulating learning plans suitable for different teaching contexts and creating knowledge connections across various disciplines. Especially in more complex interdisciplinary teaching environments, STEM lecturers increasingly need self-directed learning strategies to enhance their competency in interdisciplinary teaching and academic research skills [18].

In the course of this development, this study identifies psychological capital (a developable positive psychological resource) as the bridge connecting STEM lecturers' self-directed learning and competency in interdisciplinary teaching. Specifically, teachers with high levels of psychological capital possess additional resources to manage their teaching tasks and learning [19,20]. The self-efficacy, hope, resilience, and optimism encompassed in teachers' psychological capital are influenced to varying degrees by their self-directed learning, thereby enhancing their competency in interdisciplinary teaching [21,22]. Teachers who maintain positive expectations for their teaching outcomes can quickly recover from setbacks in teaching competency,

exhibiting high levels of optimism even under adverse circumstances [23]. Studies have also shown that teachers with high levels of psychological capital tend to view various pressures as challenges rather than threats and possess enhanced coping abilities and regulatory strategies to manage and address stress [24]. In real interdisciplinary teaching environments, STEM lecturers are increasingly required to emphasize and teach undergraduates how to regulate their psychological capital to better learn interdisciplinary knowledge and skills [25]. However, there has been no conscious training or intervention aimed at improving the psychological capital of the lecturers themselves, indicating that the role of STEM lecturers' psychological capital is underestimated and overlooked [26,27]. Therefore, in the known relationship between self-directed learning and teaching competency, focusing on the benefits brought by these positive psychological traits can help STEM lecturers perform better when facing complex and demanding interdisciplinary teaching practices. Therefore, it is crucial to outline the following research questions, which have been derived from the identified gaps in the current literature:

RQ1. Do SDL have a direct impact on the CIT among university STEM lecturers in China?

RQ2. Do SDL have a direct impact on the PsyCap among university STEM lecturers in China?

RQ3. Do PsyCap have a direct impact on the CIT among university STEM lecturers in China?

RQ4. Does PsyCap mediate the relationship between SDL and CIT among university STEM lecturers in China?

## Literature review

### Theoretical foundations

The theoretical foundation of this study is anchored in Psychological Capital Theory [28], the iceberg competency model [29], and the theory of adult learning [30] to elucidate how STEM lecturers autonomously enhance competency in interdisciplinary teaching.

SDL stems from the theory of adult learning, which emphasizes the autonomy, experiential basis, intrinsic motivation, and practical orientation of adult learners [31]. Adult learners tend to manage their learning processes actively, including setting learning goals, selecting learning resources, and evaluating learning outcomes. This theory explains how adult learners, through self-directed learning, can tailor their education to their interests and needs, rather than passively receiving instruction, thereby promoting their personal and professional development.

Psychological capital theory elucidates the role of positive psychological resources in job performance, organizational behavior, and personal growth [32]. Rooted in positive psychology and positive organizational behavior, this theory underscores the importance of positive psychological resources. It defines four core dimensions of psychological capital: self-efficacy, hope, resilience, and optimism, explaining how these dimensions synergistically enhance individual performance and well-being. Both the theory of adult learning and the theory of psychological capital in the field of education aim to enhance the learning and teaching performance of students and teachers, with teaching competency being the most crucial indicator of teaching performance.

The iceberg competency model provides a framework for understanding and developing teachers' competency in interdisciplinary teaching. This model uses the iceberg metaphor to distinguish between visible and hidden competencies [33]. Visible competencies include knowledge and skills, which are easily observed and measured, while hidden competencies encompass self-concept, traits, and motives, which are less apparent but profoundly impact performance. Through this model, organizations and individuals can better identify and develop the necessary competencies to enhance overall performance and competitiveness. Consequently, this study endeavors to elucidate a cohesive framework that interweaves self-directed learning, competency in interdisciplinary teaching and psychological capital.

 

## Conceptual framework

Guided by theoretical foundations, this study proposes a conceptual framework centered on three core constructs: self-directed learning (SDL), psychological capital (PsyCap), and competency in interdisciplinary teaching (CIT). The model aims to provide an integrated explanation of the development of interdisciplinary teaching competency among STEM university lecturers from cognitive, psychological, and behavioral perspectives.

SDL, conceptualized as a second-order construct, comprises three primary dimensions: self-management, learning desire, and self-control [34]. This learning process not only facilitates continuous improvement in lecturers' knowledge and instructional skills but also helps activate their positive psychological resources. PsyCap, which consists of four core dimensions including self-efficacy, hope, optimism, and resilience, is a developable and positively oriented psychological resource that strengthens lecturers' adaptability and sustained engagement in teaching practices [35]. CIT reflects lecturers' ability to integrate knowledge across disciplines and to design and implement cross-domain instructional activities, encompassing both explicit instructional skills and implicit psychological traits [10]. Fig 1 presents the conceptual framework, featuring SDL, PsyCap, and CIT as second-order constructs, and illustrating the hypothesized relationships among them.

It should be noted that this study does not incorporate moderating variables in this model, as existing evidence of moderation effects has been found only in the path between the learning desire dimension of SDL and outcomes such as teaching engagement or self-efficacy, where demographic characteristics such as teaching experience, professional title, and educational background may exert some influence. However, there is currently insufficient theoretical and empirical support for moderation effects in the primary paths from overall SDL to PsyCap and CIT. To avoid overcomplicating the model and weakening the clarity of path explanations, this study focuses on the mediating role of PsyCap, aiming to clarify its critical function in the relationship between self-directed learning and competency in interdisciplinary teaching, and to provide a theoretical foundation for future research involving multilevel structures or interaction effects.

## Hypotheses development

**Self-directed learning and competency in interdisciplinary teaching.** Teachers' self-directed learning emphasizes the proactive exploration and acquisition of new knowledge or skills to enhance personal teaching practice [36]. It involves teachers setting learning goals for self-management, self-assessing progress to diagnose their learning desire, and reflecting on how learning outcomes can be applied in teaching to achieve self-control. Studies have shown that a lack of self-directed learning ability among teachers can directly affect their teaching effectiveness [37]. Empirical research

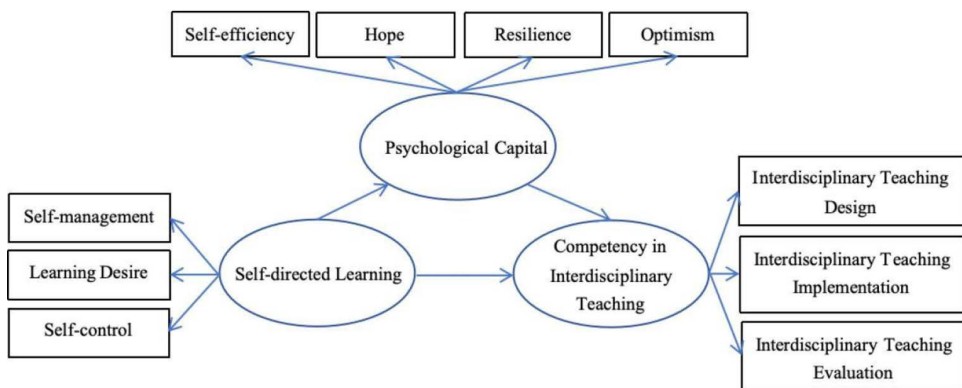

**Fig 1. Conceptual framework.**

also indicates that teachers' self-directed learning significantly influences the development of their teaching competency [38]. Teachers with higher levels of self-directed learning ability often exhibit higher levels of teaching competency [39]. For instance, research has found that K-12 teachers who independently learn through social media can enhance their teaching competencies [40]. Another study found that using a Montessori approach-based STEM program developed pre-school teacher candidates' self-directed learning skills, which in turn enhanced their teaching competency [41]. In the context of higher education, studies from the UK and Australia also confirm that university instructors who actively engage in self-directed learning tend to demonstrate greater instructional adaptability and interdisciplinary teaching responsiveness [42,43]. Furthermore, Morris found that SDL, conceptualized as a meta-competence, was positively associated with teachers' capacity to integrate and apply knowledge across disciplinary boundaries, highlighting its relevance for interdisciplinary teaching performance [16].

Although there is no direct empirical evidence indicating that self-directed learning among STEM lecturers affects their competency in interdisciplinary teaching, existing theoretical evidence suggests that STEM lecturers require more interdisciplinary knowledge and teaching skills compared to lecturers in other fields to achieve the goals of interdisciplinary teaching [44]. Moreover, competency in interdisciplinary teaching is a crucial component of overall teaching competency [39]. The degree of their engagement in self-directed learning can influence the integration and application of interdisciplinary teaching knowledge, thereby enhancing their competency in interdisciplinary teaching [45,46]. In summary, this study proposes the following hypothesis: Hypothesis 1 (H1): Self-directed learning exerts a significant and positive direct influence on competency in interdisciplinary teaching.

**Psychological capital as potential mediating factor.** Psychological capital refers to a positive psychological state that individuals possess, which serves as a developable resource to enhance individual job performance and organizational competitiveness [47]. Key components of psychological capital include self-efficacy, hope, optimism, and resilience, all of which are closely related to self-directed learning abilities [48]. Existing research has confirmed a bidirectional influence between self-directed learning and psychological capital [49]. However, this study primarily focuses on the impact of teachers' self-directed learning on psychological capital. A study found that STEM teachers who set their own learning tasks in informal learning environments became more engaged and focused. Successfully completing these tasks boosted their self-efficacy, which in turn further enhanced their motivation to learn and teaching competencies [50]. Another study revealed that during the pandemic, STEM lecturers who actively sought learning resources and formulated solutions helped maintain their sense of hope and optimism about their future teaching careers [51]. In a European university context, similar patterns were observed among higher education faculty who demonstrated increased resilience and psychological adaptability through structured SDL engagement [52]. Empirical findings also suggest that lecturers' self-directed learning attitudes significantly predict higher levels of resilience and classroom satisfaction, indicating a positive influence of SDL on psychological well-being and teaching-related emotional outcomes [53]. Hence, the presented evidence supports the following hypothesis: Hypothesis 2 (H2): Self-directed learning has a significantly positive impact on psychological capital.

Based on the aforementioned assumptions, further research indicates that teachers' psychological capital significantly influences their engagement in teaching, participation in various professional development activities, and teaching competency [54]. A large-scale survey study in China conducted a correlation analysis and moderation mediation effect test on the data regarding teachers' psychological resilience and online teaching competence during the pandemic. It states that psychological resilience positively impacted online teaching competency [19]. Similarly, studies have shown that optimism and resilience help teachers maintain a positive mindset when facing interdisciplinary teaching challenges, thereby enhancing their interdisciplinary teaching competency and student satisfaction [55]. Moreover, research on German STEM secondary school teachers found a positive relationship between self-efficacy and teaching competency in interdisciplinary science instruction [56]. In addition, empirical studies conducted in Canadian and Norway universities found that higher levels of PsyCap significantly predicted instructional effectiveness across multiple teaching domains, including

interdisciplinary settings, supporting its positive role in enhancing teaching competency [57,58]. In summary, this study proposes the third hypothesis: Hypothesis 3 (H3). Psychological capital has a significantly positive effect on competency in interdisciplinary teaching.

In the discussion above, this study first demonstrates a positive correlation between lecturers' self-directed learning and competency in interdisciplinary teaching. It then explores evidence that self-directed learning positively predicts psychological capital, which in turn affects competency in interdisciplinary teaching. Although the current literature does not directly link these three variables and lacks studies examining the relationships between self-directed learning, psychological capital, and competency in interdisciplinary teaching, existing evidence and conceptual foundations suggest that psychological capital plays a potential mediating role. Thus, the hypotheses were proposed as follows: Hypothesis 4 (H4). Psychological capital serves as a mediator in the relationship between self-directed learning and competency in interdisciplinary teaching.

## Methodology

### Population, sample and data collection

The target population of this study comprises STEM lecturers from seven first-tier public universities in Yunnan Province, China. This region is characterized by its complex borderlines, significant cross-border trade volume, and diverse cultural integration, leading to intricate socio-scientific issues. Therefore, there is a critical need for university STEM lecturers to cultivate high-quality talents with interdisciplinary skills. According to the China Educational Statistics Yearbook [12], there are 10,564 STEM lecturers across these seven universities. Using the calculator.net software, the required sample size was determined to be 371. To account for potential non-responses, an additional 15% was included as a reserve sample, resulting in a final target sample size of 427. Data were collected through an online questionnaire survey, employing a proportional stratified random sampling method to ensure sample representativeness. With the assistance of the Yunnan Provincial Department of Education, the questionnaire link, which included a QR code and a written informed consent form, was distributed to the respondents via email notifications. Although this study relied solely on self-reported data, efforts were made to reduce potential response bias, particularly social desirability bias, by assuring participants of anonymity and emphasizing that there were no right or wrong answers. Lecturers were given a three-week period (from December 15, 2024, to January 4, 2025) to complete the questionnaire at their convenience. This time was strategically chosen, as it coincided with the period after the completion of their teaching tasks, ensuring that participation would not interfere with their daily teaching and academic responsibilities. This approach was expected to enhance the quality of the collected data. A total of 427 questionnaires were distributed, with 416 returned, yielding a response rate of 97.4%. After excluding eight invalid responses, 408 valid questionnaires were retained for statistical analysis. Given the complexity of the proposed research model, this sample size was deemed sufficient, and Structural Equation Modeling (SEM) was employed for data analysis.

### Measurements

Exogenous latent construct. This study uses the questionnaire developed by Fisher and King [34]. It consists of 29 items, encompassing three dimensions: self-management, learning desire, and self-control. Each item was assessed using a five-point Likert scale, with responses ranging from 1 (Strongly Disagree) to 5 (Strongly Agree). Higher scores indicate higher levels of self-directed learning among STEM lecturers. The Cronbach's α coefficient for the questionnaire is 0.892.

Endogenous latent construct. This study uses the questionnaire competency in interdisciplinary teaching of developed by Liang and Li [10]. It consists of 26 items, covering three dimensions: ability of interdisciplinary teaching design, teaching, and evaluation. Responses to the items were measured on a five-point Likert scale, spanning from 1 (Strongly Disagree) to 5 (Strongly Agree). Higher scores indicate higher levels of competency in interdisciplinary teaching. The Cronbach's α coefficient for the questionnaire is 0.970.

Psychological capital. This study uses the questionnaire developed by Lorenz and his team [35]. It comprises 12 items, including four dimensions: self-efficiency, hope, optimism, and resilience. Responses to the items were measured on a six-point Likert scale, spanning from 1 (Strongly Disagree) to 6 (Strongly Agree). Higher scores indicate higher levels of psychological capital. The Cronbach's α coefficient for the questionnaire is 0.820.

## Data analysis

### Assessment of common method biases

To examine potential common method bias (CMB) in this study, the full collinearity test was applied using the variance inflation factor (VIF) approach recommended by Kock [59], which is particularly suitable for PLS-SEM analysis. Compared to conventional techniques such as Harman's single-factor test, the VIF method provides a more statistically robust assessment in variance-based models. The VIF values in this study ranged from 1.628 to 2.600, all remaining well below the recommended threshold of 3.3, suggesting that common method variance is not a serious concern in this research. In addition, these VIF values also indicate that multicollinearity among latent constructs is not a threat, supporting the discriminant validity of the measurement model.

### Descriptive statistics and correlation analysis

This study utilized SPSS.28 to examine the normality of the data using the Kolmogorov-Smirnov (K-S) test. The analysis indicated that none of the variables followed a normal distribution. Subsequently, descriptive statistics for the sample data were obtained, as shown in Table 1, and the correlation coefficients of the variables are presented in Table 2. The results of the Spearman correlation analysis indicated a significant positive correlation between self-directed learning and competency in interdisciplinary teaching ($r=0.574$, $p<0.001$). Additionally, self-directed learning was positively associated with psychological capital ($r=0.609$, $p<0.001$). Furthermore, psychological capital demonstrated a strong positive correlation with competency in interdisciplinary teaching ($r=0.715$, $p<0.001$).

In this study, the learning desire dimension within the variable of self-directed learning exhibited the strongest correlation with the dependent variable competency in interdisciplinary teaching ($r=0.630$, $p<0.001$) and also demonstrated a significant positive correlation with the interdisciplinary teaching evaluation dimension ($r=0.631$, $p<0.001$). Similarly, the learning desire dimension of SDL shows the strongest correlation with PsyCap ($r=0.624$, $p<0.001$), and it is also significantly correlated with the self-efficacy dimension of PsyCap ($r=0.623$, $p<0.001$). In addition, the self-efficacy dimension of psychological capital showed the highest correlation with competency in interdisciplinary teaching ($r=0.695$, $p<0.001$) and maintained a strong correlation with the interdisciplinary teaching evaluation dimension ($r=0.680$, $p<0.001$).

### Measurement model assessment

To ensure that all latent constructs exhibited robust psychometric properties, this study adopted a second-order reflective model structure, in which the three core constructs including SDL, PsyCap, and CIT were each composed of multiple first-order reflective dimensions. Subsequently, a series of validity and reliability tests were conducted. First, convergent validity was evaluated using factor loadings. The model initially comprised 67 indicators. As shown in Fig 2, 64 factor loadings exceeded the threshold of 0.7, while three indicators (LD5, SM2, and SC2) with loadings below 0.7 were removed, confirming excellent convergent validity [60]. Then, Average Variance Extracted (AVE) was also used to assess convergent validity, which reflects the ability of the latent variable to explain its observed indicators. The results in Table 3 confirm that all AVE values exceed 0.5, thereby meeting the required threshold. Based on convergent validity, discriminant validity can be assessed using two methods: the Fornell-Larcker criterion and the Heterotrait-Monotrait (HTMT) criterion. However, this study considers that the construct of learning desire in SDL and the construct of self-efficiency in PsyCap may be related in the learning context. Therefore, the HTMT criterion was used to further assess discriminant validity, as it

**Table 1. Demographic characteristics of the sample and descriptive statistics.**

| | N | % | Variables | M | SD |
|---|---|---|---|---|---|
| **Gender** | | | SDL | 3.777 | 0.647 |
| Male | 208 | 51 | | | |
| Female | 200 | 49 | SM | 3.777 | 0.919 |
| **Age** | | | | | |
| <30 | 25 | 6.1 | LD | 3.670 | 0.919 |
| 30-35 | 175 | 42.9 | | | |
| 36-40 | 166 | 40.7 | SC | 3.882 | 0.886 |
| >40 | 42 | 10.3 | | | |
| **Teaching years** | | | CIT | 3.774 | 0.717 |
| 1-5 | 106 | 26 | | | |
| 6-15 | 248 | 60.8 | ITD | 3.752 | 0.990 |
| >15 | 54 | | | | |
| **Educational background** | | | ITI | 3.879 | 0.919 |
| Bachelor degree | 31 | 7.6 | | | |
| Master Degree | 212 | 52 | ITE | 3.690 | 0.971 |
| Ph.D Degree | 165 | 40.4 | | | |
| **Professional title** | | | PsyCap | 4.293 | 0.928 |
| Teaching assistant | 71 | 17.4 | | | |
| Lecturer | 120 | 29.4 | Optimism | 4.361 | 1.236 |
| Associate professor | 141 | 34.6 | | | |
| Professor | 76 | 18.6 | Hope | 4.279 | 1.273 |
| **Subject you teach matches your highest degree** | | | | | |
| Yes | 347 | 85.04 | Resilience | 4.298 | 1.230 |
| No | 61 | 14.95 | | | |
| | | | SE | 4.235 | 1.272 |

Note: SDL = Self-directed learning, SM = Self-management, LD = Learning desire, SC = Self-control, CIT = Competency in interdisciplinary teaching, ITD = Interdisciplinary teaching design, ITI = Interdisciplinary teaching implementation, ITE = Interdisciplinary teaching evaluation, SE = Self-efficacy, M = Mean, SD = Standard deviation.

is more sensitive in detecting discriminant validity issues that the Fornell-Larcker criterion may overlook, especially when constructs are closely related but not identical [61]. The results in Table 3 show that all HTMT values were below the recommended threshold of 0.90 (or 0.85), confirming satisfactory discriminant validity.

Secondly, The reliability of the constructs was assessed using Cronbach's Alpha coefficient and Composite Reliability (CR), the latter serving as a measure of the shared variance among observed variables representing a latent construct. The results in Table 3 indicate that both Cronbach's Alpha and Composite Reliability values exceed the threshold of 0.7, thereby fulfilling the reliability requirements.

## Structural model assessment

Following the completion of the required assessment of the measurement model, the analysis proceeded to the second phase, which involved evaluating the structural model. The hypothesis testing process was carried out in several steps. First, the direct effects outlined in the model were examined. Next, 5,000 bootstrapping resamples were utilized to calculate the standard errors and assess the significance of the direct paths. Lastly, the predictive capability of the model was analyzed using the Blindfolding algorithm.

**Table 2. Correlations of the study variables (N = 408).**

| Variables | 1 | 2 | 3 | 4 | 5 | 6 | 7 | 8 | 9 | 10 | 11 | 12 | 13 |
|---|---|---|---|---|---|---|---|---|---|---|---|---|---|
| 1. SDL | 1 | | | | | | | | | | | | |
| 2. SM | .746*** | 1 | | | | | | | | | | | |
| 3. LD | .773*** | .451*** | 1 | | | | | | | | | | |
| 4. SC | .742*** | .427*** | .489*** | 1 | | | | | | | | | |
| 5. CIT | .574*** | .443*** | .630*** | .462*** | 1 | | | | | | | | |
| 6. ITD | .507*** | .433*** | .583*** | .417*** | .852*** | 1 | | | | | | | |
| 7. ITI | .518*** | .406*** | .556*** | .468*** | .795*** | .561*** | 1 | | | | | | |
| 8. ITE | .570*** | .440*** | .631*** | .460*** | .845*** | .687*** | .585*** | 1 | | | | | |
| 9. PsyCap | .609*** | .429*** | .642*** | .489*** | .715*** | .647*** | .578*** | .676*** | 1 | | | | |
| 10. Optimism | .528*** | .341*** | .619*** | .414*** | .691*** | .640*** | .543*** | .705*** | .813*** | 1 | | | |
| 11. Hope | .466*** | .442*** | .412*** | .404*** | .451*** | .394*** | .432*** | .392*** | .693*** | .404*** | 1 | | |
| 12. Resilience | .575*** | .397*** | .613*** | .487*** | .691*** | .621*** | .556*** | .691*** | .843*** | .699*** | .453*** | 1 | |
| 13. SE | .551*** | .386*** | .623*** | .429*** | .695*** | .659*** | .570*** | .680*** | .861*** | .713*** | .480*** | .706*** | 1 |

Note:

***p < 0.001.

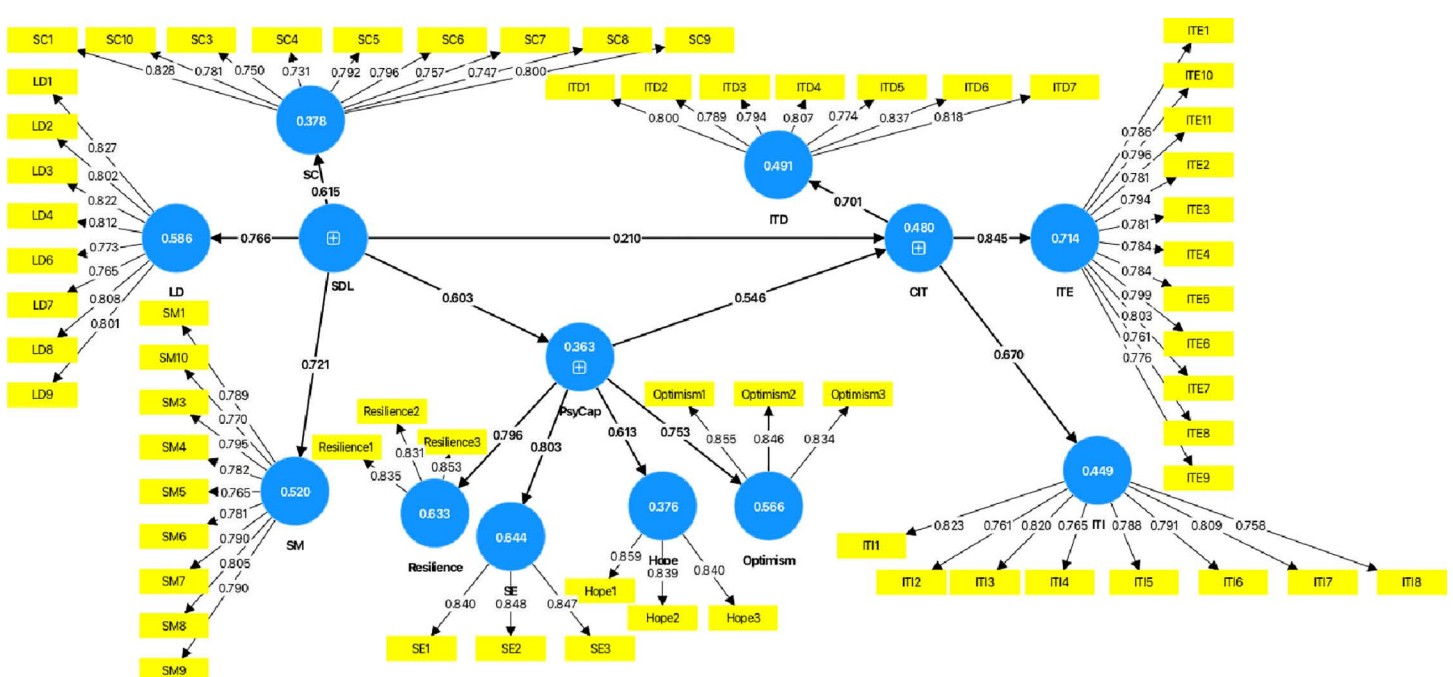

**Fig 2. The SEM diagram depicts the connections between SDL, CIT, and PsyCap.**

Table 4 and Fig 2 demonstrates a significant relationship between SDL and CIT (β = 0.210, p < 0.001), with the 95% confidence interval excluding 0, thus confirming the acceptance of H1. Additionally, the direct effect of SDL on PsyCap (β = 0.603, p < 0.001) was also found to be significant, with the 95% confidence interval excluding 0, thereby supporting H2. Similarly, the direct effect of PsyCap on CIT (β = 0.546, p < 0.001), with the 95% confidence interval excluding 0, was revealed to be significantly positive, leading to the acceptance of H3.

**Table 3. The reliability and validity testing of the model.**

| Variables | α | CR | AVE | HTMT |
|---|---|---|---|---|
| 1. SDL | 0.911 | 0.920 | | SDL <-> CIT, 0.555 |
| 2. SM | 0.922 | 0.935 | 0.617 | |
| 3. LD | 0.920 | 0.935 | 0.643 | |
| 4. SC | 0.917 | 0.932 | 0.603 | |
| 5. CIT | 0.927 | 0.935 | | PsyCap <-> CIT, 0.660 |
| 6. ITD | 0.908 | 0.927 | 0.645 | |
| 7. ITI | 0.914 | 0.930 | 0.624 | |
| 8. ITE | 0.938 | 0.947 | 0.618 | |
| 9. PsyCap | 0.859 | 0.886 | | SDL <-> PsyCap, 0.746 |
| 10. Optimism | 0.800 | 0.882 | 0.714 | |
| 11. Hope | 0.802 | 0.883 | 0.716 | |
| 12. Resilience | 0.791 | 0.878 | 0.705 | |
| 13. SE | 0.800 | 0.882 | 0.714 | |

**Table 4. Structural model path coefficient results (direct effects).**

| Hypotheses | Relationship | β | $R^2$ | $f^2$ | 95%CI | t-Value | Decision |
|---|---|---|---|---|---|---|---|
| H1 | SDL→CIT | 0.210 | 0.478 | 0.054 | [0.119, 0.304] | 4.429 | Supported |
| H2 | SDL→PsyCap | 0.603 | 0.362 | 0.570 | [0.531, 0.672] | 16.737 | Supported |
| H3 | PsyCap→CIT | 0.546 | 0.478 | 0.365 | [0.461, 0.629] | 12.748 | Supported |

In terms of explained variance, the $R^2$ for PsyCap was 0.362 ($p < 0.001$), indicating that SDL accounts for 36.2% of the variance in PsyCap. The $R^2$ for CIT was 0.478 ($p < 0.001$), meaning that SDL and PsyCap together explain 47.8% of the variance in CIT.

The $f^2$ values for the effects of SDL on CIT, SDL on PsyCap, and PsyCap on CIT were 0.054, 0.570, and 0.365, respectively. According to Cohen's criteria (where an $f^2$ value of approximately 0.02 represents a small effect, around 0.15 represents a medium effect, and around 0.35 indicates a large effect), It is evident that SDL has a large predictive effect on PsyCap, and PsyCap, in turn, exerts large predictive effect on CIT, whereas the predictive effect of SDL on CIT is relatively small.

Additionally, the model demonstrated an acceptable overall fit, as evidenced by the following indices: SRMR = 0.051, d_ULS = 0.550, d_G = 0.185, and NFI = 0.906 (see Table 5 below). Based on this satisfactory model fit, further evaluation of predictive relevance was conducted using the Blindfolding procedure in SmartPLS 4.0. The omission distance (D) was set to 7, in accordance with the recommended criterion that the sample size (N = 408) should not be divisible by D. The $Q^2$ values for SDL (0.037), PsyCap (0.141), and CIT (0.169) were all greater than zero, indicating that the model has adequate predictive relevance.

**Table 5. Model fit indices and recommended thresholds.**

| Fit index | SRMR | D_ULS | D_G | NFI |
|---|---|---|---|---|
| Research model | 0.074 | 0.807 | 0.165 | 0.706 |
| Recommended criteria | <0.08 | >0.05 | >0.05 | >0.09 |

## Mediation analysis

To verify whether PsyCap mediates the relationship between SDL and CIT, this study employed partial least squares structural equation modeling (PLS-SEM) to analyze the indirect, direct, and total effects between variables. The mediation analysis in PLS-SEM was conducted using the bootstrapping method with 5,000 resamples. The results reveal that the indirect effect of SDL on CIT via PsyCap is 0.329 (P<0.001), reaching a significant level, which indicates the presence of a mediation effect. Additionally, the direct effect of SDL on CIT is 0.210 (P<0.001), also significant, suggesting that PsyCap serves as a partial mediator between SDL and CIT. Given that both the direct and indirect effects are positive, this mediation is classified as complementary mediation. Thus, hypothesis H4 is supported (see Table 6).

## Robustness checks for structural model assessment

After the structural model was established, robustness checks were conducted following the recommendations of Hair [62] and Sarstedt [63], focusing on three key aspects: nonlinearity, endogeneity, and unobserved heterogeneity. These additional assessments were undertaken to further evaluate the explanatory validity and statistical stability of the model.

First, to examine the potential presence of nonlinear relationships among latent constructs, the quadratic effect approach was applied in SmartPLS, as suggested by Hair [62]. Specifically, quadratic terms of SDL and PsyCap were included to extend the path analysis. As presented in Table 7, the nonlinear effect from SDL to PsyCap was statistically significant, as was the nonlinear effect from PsyCap to CIT, indicating the presence of curvilinear mechanisms along the mediating path. In contrast, the quadratic term for the direct path from SDL to CIT was not significant, supporting the appropriateness of linear specification for this path. These results suggest that certain relationships in the model exhibit nonlinear characteristics, which help to uncover more complex structural associations between constructs and enhance both theoretical interpretability and model robustness.

Second, in accordance with Sarstedt [63], the Gaussian Copula method was employed to assess potential endogeneity concerns. This technique identifies whether the predictor variables are correlated with the structural error terms, thereby evaluating whether path estimates are biased. As shown in Table 7, none of the key paths demonstrated statistically significant p-values, indicating that no substantial endogeneity issue exists among the core constructs. Therefore, the structural path estimates can be considered robust and reliable.

Finally, the study further assessed unobserved heterogeneity in the sample. Following the recommendation of Becker [64], the Finite Mixture Partial Least Squares (FIMIX PLS) method was employed to identify potential latent segments and determine whether hidden subgroup structures exist in the data. As shown in Table 8, when the number of segments was set to three, key model selection criteria such as AIC, BIC, CAIC, and HQ demonstrated relatively lower values compared

**Table 6. Indirect effect, direct effect and total effect.**

| Hypotheses | Relationship | Direct effect | Indirect effect | Total effect | t-Value | Decision |
|---|---|---|---|---|---|---|
| H4 | SDL→PsyCap→CIT | 0.210 | 0.329 | 0.539 | 9.738 | Supported |

**Table 7. The result of Quadratic effect and Gaussian copula.**

| Relationship | Quadratic effect | | Gaussian copula | |
|---|---|---|---|---|
| | T | P | T | P |
| SDL→CIT | 0.687 | 0.492 | 1.374 | 0.189 |
| SDL→PsyCap | 4.254 | 0.000 | 1.442 | 0.152 |
| PsyCap→CIT | 2.595 | 0.009 | 1.218 | 0.223 |

**Table 8. The result of unobserved heterogeneity.**

| Criteria | Segment 2 | Segment 3 | Segment 4 |
|---|---|---|---|
| AIC (Akaike's information criterion) | 7082.048 | 6429.734 | 5989.229 |
| AIC3 (modified AIC with factor 3) | 7133.048 | 6506.743 | 6092.229 |
| AIC4 (modified AIC with factor 4) | 7184.048 | 6583.743 | 6195.229 |
| BIC (Bayesian information criterion) | 7286.623 | 6738.611 | 6402.390 |
| CAIC (consistent AIC) | 7337.623 | 6815.611 | 6505.390 |
| HQ (Hannan-Quinn criterion) | 7162.99 | 6551.963 | 6152.718 |
| MDL5 (minimum description length with factor 5) | 8512.921 | 8590.081 | 8879.032 |
| LnL (LogLikelihood) | −3490.024 | −3137.872 | −2891.615 |
| EN (normed entropy statistic) | 0.994 | 0.983 | 0.955 |
| NFI (non-fuzzy index) | 0.996 | 0.985 | 0.954 |
| NEC (normalized entropy criterion) | 2.271 | 6.968 | 18.301 |

to the two-or four-segment solutions, suggesting better explanatory performance. Moreover, the NEC value reached 6.968 under the three-segment solution, which indicates an acceptable level of heterogeneity. These results imply the possible presence of latent subpopulations within the dataset. Future research may consider conducting group comparisons or multi-group analyses to further explore this structure.

## Discussion, conclusion and implication

### Discussion

Based on a comprehensive analysis of the literature, validation of the measurement and structural models, and mediation analysis, this study is the first to confirm the hypothesis that SDL significantly and positively predicts CIT. This underscores the critical importance of SDL for university STEM lecturers, who must balance the dual roles of teaching and learning, in enhancing their CIT. Although no studies until now have reported this finding, prior research predominantly focuses on the relationship between SDL in students and teaching competency in educators [38,65]. These studies often treat teaching and learning as separate entities, neglecting the interconnected nature of these processes within educators themselves. Most investigations have only considered either the teaching role of teachers or the learning role of students, without empirically integrating the dual roles that teachers embody in both teaching and learning [66–68]. Contrary to these studies, this research empirically integrates both roles within STEM lecturers, providing a more holistic understanding of their interdisciplinary teaching development. Moreover, existing research has largely centered on teaching competency, offering limited insights into how SDL can be leveraged by university STEM lecturers to independently enhance their CIT [18,69,70]. This study uniquely positions SDL as an autonomous driver for CIT specifically within STEM higher education contexts, thereby extending adult learning theory and the iceberg competency model into interdisciplinary teaching scenarios and broadening the theoretical and empirical scope of research in this area.

In addition, the hypothesis that SDL significantly and positively predicts PsyCap was confirmed, which strengthened and expanded the application scope of PsyCap in higher education. This finding aligns with previous research, which also observed a strong correlation between SDL and key components of PsyCap such as resilience and self-efficacy among medical students and STEM educators [71–73]. Compared to these prior studies, our research highlights SDL's role in sustaining internal psychological balance when STEM lecturers face intensive teaching demands, offering a new perspective on how learning autonomy contributes to psychological resilience. Further, this research was the first to explore the beneficial roles of PsyCap in enhancing both SDL and CIT simultaneously. PsyCap significantly predicts CIT, with self-efficacy being the most influential dimension. This outcome is consistent with psychological capital theory, which posits that individuals with higher levels of PsyCap perform better in their professional roles [24,58,74,75]. Whereas prior studies

mainly explored PsyCap's relationship with job satisfaction or motivation, our study demonstrates its direct predictive role in shaping competency in interdisciplinary teaching, an underexplored area in existing PsyCap research. Our research also provided empirical evidence to differentiate the roles that PsyCap dimensions play in enhancing teaching competency. The findings support existing evidence that higher levels of resilience enable educators to better manage complex teaching tasks and improve their teaching competency [76]. By focusing on STEM lecturers, who face significant interdisciplinary teaching pressures, this study enriches the literature by unveiling the pathway effect between PsyCap and CIT, further strengthening the empirical evidence linking PsyCap to teaching competency.

Finally, this study has certain limitations that may inform future research directions and enhance the applicability of the findings. One notable concern is that the relationships among variables may not be strictly linear. The results revealed nonlinear effects from SDL to Psycap, and from Psycap to CIT, suggesting that more complex mechanisms may underlie these associations. As the structural model was developed based on linear assumptions, it may not have fully captured such nonlinear patterns, which could partly explain the moderate level of explanatory power for CIT ($R^2 = 0.362$). In addition, the study employed a cross-sectional design, and the sample was limited to STEM lecturers from selected regions in China. These contextual and institutional constraints may limit the generalizability of the findings. Moreover, the study relied solely on self-reported questionnaire data without triangulation, which may affect the methodological robustness. Future studies are encouraged to adopt longitudinal designs and nonlinear modeling techniques, as well as to expand the sample scope to more comprehensively uncover the actual relationships among the variables. In addition, as this study did not include any moderating variables, future research may consider incorporating moderation mechanisms and extending the current framework to a moderated mediation model, thereby deepening the understanding of how SDL influences CIT through PsyCap.

## Implication

The theoretical contribution of this study lies in its novel integration of SDL, PsyCap and CIT into a unified conceptual framework that systematically explores the mechanisms linking these constructs. Previous research has typically examined the impact of SDL on teaching competency, the role of SDL in enhancing PsyCap, or the effect of PsyCap on teaching competence within varying contexts. Existing research has not yet recognized that CIT is a subordinate construct within teaching competency, specifically reflecting the teaching competency that teachers demonstrate when carrying out interdisciplinary teaching tasks [10]. By synthesizing these three variables, this study reveals PsyCap as a complementary mediator between SDL and CIT, offering a new theoretical perspective on the mechanisms driving CIT enhancement among university STEM lecturers. This framework also broadens the application of adult learning theory, psychological capital theory, and the iceberg competency model.

The practical implications of this study are multifaceted. Firstly, findings indicate that SDL significantly enhances CIT among STEM lecturers in higher education. Universities are advised to develop structured SDL programs within their faculty development centers, offering modular, goal-setting-based workshops tailored to interdisciplinary teaching contexts. These programs could incorporate digital learning portfolios to track lecturers' progress, and provide micro-credentials upon completion to incentivize participation and recognize professional growth. Secondly, this study suggests that SDL facilitates the development of PsyCap. To further reinforce STEM lecturers' resilience and self-efficacy, institutional leaders may integrate structured reflection practices into existing performance review systems. Lecturers could be encouraged to maintain SDL journals or digital learning logs, which may serve as evidence in annual evaluations or teaching award nominations. In addition, resilience training programs could be implemented to help lecturers manage stress and adapt to the demands of interdisciplinary teaching more effectively. Lastly, this study highlights the positive impact of psychological capital, particularly self-efficacy, resilience, and optimism, on interdisciplinary teaching competency. Accordingly, it is recommended that universities are encouraged to establish PsyCap support schemes coordinated jointly by counseling centers and faculty affairs offices, including semester-based resilience workshops, peer coaching groups, and targeted

counseling sessions focusing on confidence-building for interdisciplinary teaching. These initiatives should be embedded in the broader professional development policies to ensure sustainability and institutional support.

## Conclusion

This study, conducted in the context of the interdisciplinary teaching challenges faced by STEM lecturers, confirms the strong measurement properties of the models for SDL, PsyCap, and CIT, offering valuable insights and a foundation for future research. The findings reveal several key relationships: (1) SDL significantly predicts CIT, with the dimension of learning desire exhibiting the strongest correlation with CIT's interdisciplinary teaching evaluation. (2) SDL also significantly predicts PsyCap, particularly with learning desire showing the strongest correlation with PsyCap's resilience. (3) PsyCap significantly predicts CIT, with self-efficacy emerging as the most strongly correlated dimension with CIT's interdisciplinary teaching evaluation. (4) PsyCap plays a complementary mediating role in the relationship between SDL and CIT, further amplifying the impact of SDL on CIT.

These findings not only validate the critical role of SDL and PsyCap in fostering CIT but also provide a robust framework for exploring how PsyCap can enhance educational outcomes. The complementary mediation effect of PsyCap highlights its importance in supporting STEM lecturers as they navigate the complexities of interdisciplinary teaching, offering a valuable pathway for future educational interventions and research.

## Acknowledgments

All the authors wish to thank all of the participants.

## Author contributions

**Conceptualization:** Wenhao Su.

**Data curation:** Wenhao Su.

**Investigation:** Wenhao Su.

**Methodology:** Wenhao Su.

**Supervision:** Hidayah Mohd Fadzil, Rose Amnah Abd. Rauf.

**Writing – original draft:** Wenhao Su.

**Writing – review & editing:** Wenhao Su.

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
