## [Decision Letter · Decision Letter 0]

Dear Dr. Fadzilb,

Thank you for submitting your manuscript to PLOS ONE. After careful consideration, we feel that it has merit but does not fully meet PLOS ONE’s publication criteria as it currently stands. Therefore, we invite you to submit a revised version of the manuscript that addresses the points raised during the review process.

This study addresses a timely and relevant issue in higher education and provides valuable insights into the role of Psychological Capital in supporting interdisciplinary teaching among STEM lecturers. However, a few revisions are suggested before the manuscript can proceed further:

**Revisions Required:**

**Abstract** :

Include statistical data of the result while stating the key findings to bring rigor and validate the research claims.Briefly mention study limitations and future research directions to emphasize the originality of the study.

**Introduction & Literature Review** :

Ensure that all citations are properly included in the reference list. Missing references such as (Liang & Li, 2023), (China Education Yearbook, 2023), (Jin, 2021), etc., should be included.Consider incorporating recent empirical studies beyond China to improve the study’s global relevance and generalizability.Address potential moderating variables (e.g., institutional support, teaching experience, or disciplinary variations) to strengthen the conceptual framework.

**Methodology & Results** :

While the PLS-SEM approach is well-applied, reporting model fit indices (e.g., SRMR, NFI) would improve the robustness of the model evaluation.The R² value for PsyCap (0.362) is relatively low. Consider reporting Adjusted R² to account for model complexity and improve predictive accuracy.As suggested by Reviewer 1, conduct robustness checks for the Structural Model Assessment [including: Linearity (Quadratic effect), Heterogeneity (FIMIX-PLS), and Endogeneity (Gaussian Copula)]. Fir this, refer to the recommended paper: Technology in Farming: Unleashing Farmers’ Behavioral Intention for the Adoption of Agriculture 5.0

**Conclusion** :

Discuss study limitations (e.g., contextual constraints, generalizability issues) more explicitly.Provide clearer policy recommendations for universities and educational institutions on implementing SDL and PsyCap programs.

Addressing the above points will significantly enhance the clarity, rigor, and impact of the study.

I appreciate your contributions to this field and look forward to receiving the revised manuscript. 

We look forward to receiving your revised manuscript.

Kind regards,

Dipendra Karki, Ph.D.

Academic Editor

PLOS ONE

Journal Requirements:

Additional Editor Comments:

Dear Author/s,

Thank you for submitting your manuscript, Bridging Self-directed Learning and Competency in Interdisciplinary Teaching Among STEM Lecturers: How Psychological Capital Connects, to PLOS ONE. Your study addresses a timely and relevant issue in higher education and provides valuable insights into the role of Psychological Capital in supporting interdisciplinary teaching among STEM lecturers. However, a few revisions are suggested before the manuscript can proceed further:

Revisions Required:

Abstract:

Include statistical data of the result while stating the key findings to bring rigor and validate the research claims.

Briefly mention study limitations and future research directions to emphasize the originality of the study.

Introduction & Literature Review:

Ensure that all citations are properly included in the reference list. Missing references such as (Liang & Li, 2023), (China Education Yearbook, 2023), (Jin, 2021), etc., should be included.

Consider incorporating recent empirical studies beyond China to improve the study’s global relevance and generalizability.

Address potential moderating variables (e.g., institutional support, teaching experience, or disciplinary variations) to strengthen the conceptual framework.

Methodology & Results:

While the PLS-SEM approach is well-applied, reporting model fit indices (e.g., SRMR, NFI) would improve the robustness of the model evaluation.

The R² value for PsyCap (0.362) is relatively low. Consider reporting Adjusted R² to account for model complexity and improve predictive accuracy.

As suggested by Reviewer 1, conduct robustness checks for the Structural Model Assessment [including: Linearity (Quadratic effect), Heterogeneity (FIMIX-PLS), and Endogeneity (Gaussian Copula)]. Fir this, refer to the recommended paper: Technology in Farming: Unleashing Farmers’ Behavioral Intention for the Adoption of Agriculture 5.0

Conclusion:

Discuss study limitations (e.g., contextual constraints, generalizability issues) more explicitly.

Provide clearer policy recommendations for universities and educational institutions on implementing SDL and PsyCap programs.

Addressing the above points will significantly enhance the clarity, rigor, and impact of the study.

I appreciate your contributions to this field and look forward to receiving the revised manuscript.

Academic Editor

PLOS One

Reviewers' comments:

Reviewer's Responses to Questions

**Comments to the Author**

1. Is the manuscript technically sound, and do the data support the conclusions?

Reviewer #1: Yes

Reviewer #2: Yes

2. Has the statistical analysis been performed appropriately and rigorously?

Reviewer #1: Yes

Reviewer #2: Yes

3. Have the authors made all data underlying the findings in their manuscript fully available?

Reviewer #1: Yes

Reviewer #2: Yes

4. Is the manuscript presented in an intelligible fashion and written in standard English?

Reviewer #1: Yes

Reviewer #2: Yes

Reviewer #1: The study addresses an important gap in STEM education, specifically the role of Psychological Capital (PsyCap) in mediating self-directed learning (SDL) and competency in interdisciplinary teaching (CIT). PLS-SEM (Partial Least Squares Structural Equation Modeling) is a strong statistical method for analyzing complex relationships. However, The study relies on questionnaires, which are subject to response bias. The authors did not use triangulation. Although the Harman’s single-factor test was performed, it is not the most rigorous test for CMB. While the study suggests SDL programs and PsyCap training, it does not offer concrete policy recommendations or implementation strategies for universities.

Reviewer #2: Dear authors,

Thank you for this paper which deals with "Bridging Self-directed Learning and Competency in Interdisciplinary Teaching Among STEM Lecturers: How Psychological Capital Connects." I have thoroughly read the paper and find the proposed model is theoretically backed, constructs well defined and items properly adopted from the valid sources. Technically, the paper is very good. Abstract is short, sweet and intact. My overall impression of paper is good.

The followings are my sectionwise comments:

1. Abstract: It is well written within words limit and convey the message of the paper nicely and clearly.

2. Introduction: Intro part is also clearly and nicely written explaining all constructs ( exogenous, endogenous and mediating). However, I suggest to add some 1-2 lines about Psychological Capital in lines between 88-95 where relevant. I know you mentioned in chapter 2 about this concept. Just ‘scratch the subject to hook your audience’ here.

3. LR part: I suggest to rewrite the section 2 as Theoretical Base, Conceptual Framework and Hypotheses Development . Just follow the technical structure of academic research : Theory � Model � Constructs � Items.

In addition, as your model consists of three constructs: Exogenous Latent (SDL), Mediator (PsyCap) and Endogenous Latent Construct (CIT) and all constructs have 3-4 different dimensions. Have you applied Higher Order Construct (HoC) or measured in the first order of all items? Clarify it. this is not clear in the manuscript.

4. Methodology: In the language of SEM, it think the words like Exogeneous Latent Construct, Endogenous Latent Construct are more technical and suitable in place of Independent and Dependent Variables. I suggest to replace properly.

5. Data Analysis:

a) Assessment of CMB: First justify the ‘Procedural Remedies’ of CMB because ‘Statistical Remedies’ are not sufficient to justify. As Harman Single factor test is considered as a weak one, you further emphasize the VIF test in PLS-SEM. I suggest you test CLF (common latent factor) method to validate the CMB issue.

Structural Model Assessment: Manuscript analyzed R2, F2 and Q2. But I strongly suggest to conduct robustness checks, particularly Linearity (Quadratic effect), Heterogeneity test (FIMIX PLS) and Endogeneity (Gaussian Copula).

For the reference read the paper: Technology in farming: Unleashing farmers’ behavioral intention for the adoption of agriculture 5.0 (https://journals.plos.org/plosone/article?id=10.1371/journal.pone.0308883)

6. Discussion: it is well written. I suggest to add more citations (comparing and contrasting with previous articles in the similar field.

7. Implication and conclusion: both section seem almost good.

8. Please check references minutely as per journal guidelines and correct them.

I believe the comments mentioned above will be helpful in further developing this manuscript.

Thank you.

**Do you want your identity to be public for this peer review?** For information about this choice, including consent withdrawal, please see our Privacy Policy

Reviewer #1: **Yes: ** Assoc.Prof.Dr.Nath Amornpinyo

Reviewer #2: **Yes: ** Devid Kumar Basyal

---

## [Author Response · Author response to Decision Letter 1]

23 Apr 2025

Dear Editors and Reviewers:

Thank you for your letter and for the reviewers’ comments concerning our manuscript entitled “Bridging self-directed learning and competency in interdisciplinary teaching among STEM lecturers: How psychological capital connects” (PONE-D-25-02654). Those comments are all valuable and very helpful for revising and improving our paper, as well as the important guiding significance to our researches. We have studied comments carefully and have made correction which we hope meet with approval. Revised portion are marked in yellow in the paper. The main corrections in the paper and the responds to the reviewer’s comments are as flowing:

Responds to the reviewer’s comments:

Reviewer #1:

1. Response to comment: The study addresses an important gap in STEM education, specifically the role of Psychological Capital (PsyCap) in mediating self-directed learning (SDL) and competency in interdisciplinary teaching (CIT). PLS-SEM (Partial Least Squares Structural Equation Modeling) is a strong statistical method for analyzing complex relationships. However, The study relies on questionnaires, which are subject to response bias. The authors did not use triangulation. Although the Harman’s single-factor test was performed, it is not the most rigorous test for CMB. While the study suggests SDL programs and PsyCap training, it does not offer concrete policy recommendations or implementation strategies for universities.

Response: Thank you for your valuable feedback. Based on your comments, we have addressed the following concerns:

(1)In response to the issue of response bias, we have added a clarification in lines 285-288 of the revised manuscript, emphasizing that respondents were assured of anonymity and informed that there were no right or wrong answers in order to minimize social desirability bias. In fact, these procedural remedies were already considered during the design of the survey, but were not explicitly stated in the initial submission due to space limitations. Additionally, the limitation of not employing triangulation has been explicitly acknowledged in lines 525-527, with a recommendation for future research to incorporate multiple data sources to enhance the validity and robustness of findings.

(2)Regarding common method bias (CMB): Although we initially used Harman’s single-factor test, we have now supplemented this with a Full Collinearity VIF test, following the recommendation of Kock (2015), to enhance the statistical robustness of our assessment. The revised explanation and results of the VIF test are provided in lines 320-329 of the manuscript.

(3)We have substantially revised the practical implications section to provide more concrete and actionable recommendations for universities and educational institutions. Specifically, we now outline how structured SDL programs can be developed through modular, goal-setting-based workshops and integrated with digital learning portfolios and micro-credential systems. We also suggest embedding PsyCap development into institutional review mechanisms through SDL journals and reflective documentation. In addition, we propose concrete support strategies, including resilience training, peer coaching, and targeted counseling services, to enhance PsyCap among STEM lecturers. These revised implications (see lines 549-553 and 562-568) are intended to offer clearer implementation pathways and support evidence-based policymaking in higher education.

Reviewer #2:

1. Response to comment: Abstract: It is well written within words limit and convey the message of the paper nicely and clearly.

Response: Thank you for your comments. In response to the editor’s suggestion, we have revised the abstract to briefly acknowledge the study’s contextual and methodological limitations, thereby aligning it more closely with the journal’s expectations for transparency and completeness.

2. Response to comment: Introduction: Intro part is also clearly and nicely written explaining all constructs ( exogenous, endogenous and mediating). However, I suggest to add some 1-2 lines about Psychological Capital in lines between 88-95 where relevant. I know you mentioned in chapter 2 about this concept. Just ‘scratch the subject to hook your audience’ here.

Response: Thank you for your constructive suggestion. In response, we have revised the introduction section and added a brief explanation of Psychological Capital in lines 87-88. This addition is intended to provide a concise conceptual cue to the readers by highlighting Psychological Capital as a developable positive psychological resource. We believe this early reference enhances conceptual clarity and serves to hook the audience’s interest in its mediating role.

3.Response to comment: LR part: I suggest to rewrite the section 2 as Theoretical Base, Conceptual Framework and Hypotheses Development . Just follow the technical structure of academic research : Theory, Model, Constructs, Items.

In addition, as your model consists of three constructs: Exogenous Latent (SDL), Mediator (PsyCap) and Endogenous Latent Construct (CIT) and all constructs have 3-4 different dimensions. Have you applied Higher Order Construct (HoC) or measured in the first order of all items? Clarify it. this is not clear in the manuscript.

Response: Thank you for your insightful recommendation. In response, we have reorganized Section 2 to follow a more structured academic format, dividing it into three sub-sections: Theoretical Foundations, Conceptual Framework, and Hypotheses Development. The Conceptual Framework section has been added and is presented in lines 152-184. This revision clarifies the theoretical logic and model structure by aligning with the “Theory-Model-Constructs-Items” sequence, thereby improving the coherence and academic rigor of the literature review.

Besides, We confirm that all three latent constructs—SDL, PsyCap, and CIT—were conceptualized and analyzed as second-order reflective constructs, each composed of multiple first-order reflective dimensions. To improve clarity, we have now explicitly stated this modeling approach in the revised manuscript (lines 359-363), within the measurement model assessment section.

4.Response to comment: Methodology: In the language of SEM, it think the words like Exogeneous Latent Construct, Endogenous Latent Construct are more technical and suitable in place of Independent and Dependent Variables. I suggest to replace properly.

Response: Thank you for pointing this out. We fully agree that in the context of Structural Equation Modeling (SEM), the use of technical terms such as Exogenous Latent Construct and Endogenous Latent Construct is more appropriate than Independent and Dependent Variables. Accordingly, we have revised the terminology throughout the Methodology section, including replacements at lines 299 and 305, to ensure consistency with SEM conventions and improve conceptual precision.

5.Response to comment: Data Analysis:

a) Assessment of CMB: First justify the ‘Procedural Remedies’ of CMB because ‘Statistical Remedies’ are not sufficient to justify. As Harman Single factor test is considered as a weak one, you further emphasize the VIF test in PLS-SEM. I suggest you test CLF (common latent factor) method to validate the CMB issue.

b) Structural Model Assessment: Manuscript analyzed R2, F2 and Q2. But I strongly suggest to conduct robustness checks, particularly Linearity (Quadratic effect), Heterogeneity test (FIMIX PLS) and Endogeneity (Gaussian Copula).

Response:

a)Thank you for highlighting this important concern. In response, We emphasized the use of the Variance Inflation Factor (VIF) test for full collinearity assessment in PLS-SEM, following the approach suggested by Kock (2015), to statistically evaluate potential common method bias (CMB). These revisions are reflected in lines 320-329. While we acknowledge the theoretical value of the Common Latent Factor (CLF) approach for detecting CMB, this method is not currently supported in SmartPLS. Therefore, it was not applied in this study, but we recognize its relevance for future research in platforms where it is implementable.

b) We sincerely appreciate your suggestion to strengthen the structural model assessment. Accordingly, we have added a new subsection titled “Robustness Checks for Structural Model Assessment” (lines 434-470), in which we conducted three additional robustness tests:

(1) Linearity check using quadratic term testing,

(2) Unobserved heterogeneity analysis using FIMIX-PLS based on information criteria (AIC, BIC, etc.), and

(3) Endogeneity test using the Gaussian Copula approach.

The results of these tests confirmed the robustness of our model and added further credibility to the findings.

6.Response to comment: Discussion: it is well written. I suggest to add more citations (comparing and contrasting with previous articles in the similar field).

Response: Thank you for your valuable suggestion. In response, we have carefully reviewed and expanded the discussion section by incorporating additional citations to strengthen the theoretical foundation and better contextualize our findings. These references were added specifically to facilitate comparison and contrast with previous empirical studies on SDL, PsyCap, and interdisciplinary teaching competency among STEM lecturers. The newly inserted content, highlighted in yellow (lines 484-486, 488-492, 497-500, 505-508), reflects these comparative insights. Additionally, as advised by the editor, we revised the limitation section to acknowledge contextual and generalizability concerns and to emphasize avenues for future research.

7.Response to comment: Please check references minutely as per journal guidelines and correct them.

Response: Thank you for your careful examination. We have thoroughly reviewed and revised all references to ensure full compliance with the journal’s formatting guidelines, including author order, punctuation, journal titles, volume/issue numbers, page ranges, and DOI formatting. All inconsistencies have been corrected accordingly.

Editor:

1.Response to comment: Abstract:

Include statistical data of the result while stating the key findings to bring rigor and validate the research claims.

Briefly mention study limitations and future research directions to emphasize the originality of the study.

Response:

Thank you for your constructive suggestions. In response, we have revised the abstract (lines 30-42) to include key statistical findings that support the study’s conclusions, thereby enhancing the rigor and credibility of the research claims. Additionally, we have briefly stated the study’s limitations and suggested future research directions to better highlight the originality and potential contributions of this work.

2.Response to comment: Introduction & Literature Review:

Ensure that all citations are properly included in the reference list. Missing references such as (Liang & Li, 2023), (China Education Yearbook, 2023), (Jin, 2021), etc., should be included.

Consider incorporating recent empirical studies beyond China to improve the study’s global relevance and generalizability.

Address potential moderating variables (e.g., institutional support, teaching experience, or disciplinary variations) to strengthen the conceptual framework.

Response:

Missing References: All previously missing citations have now been properly included in the reference list and verified for accuracy.

Recent International Empirical Studies: To enhance the global relevance and generalizability of our study, we have incorporated additional empirical literature from outside China. These include recent studies conducted in European, Australian, and Canadian contexts (see lines 200-206, 232-238, and 250-256), which strengthen the cross-national applicability of our arguments regarding SDL, PsyCap, and competency in interdisciplinary teaching.

Moderating Variables in the Conceptual Framework: While we did not include moderating variables in the structural model, we have provided a theoretical rationale for this decision in the conceptual framework section (lines 173-184). Specifically, we explained that existing moderation effects have been observed only in certain sub-dimensions of SDL (e.g., learning desire), and primarily in relation to demographic variables such as teaching experience or professional title. However, there is currently insufficient theoretical and empirical evidence to justify moderation in the primary paths from overall SDL to PsyCap and CIT. To maintain model clarity and parsimony, this study focuses on the mediating role of PsyCap, while suggesting the exploration of potential moderation effects as a future research direction.

3.Response to comment: Methodology & Results:

While the PLS-SEM approach is well-applied, reporting model fit indices (e.g., SRMR, NFI) would improve the robustness of the model evaluation.

The R² value for PsyCap (0.362) is relatively low. Consider reporting Adjusted R² to account for model complexity and improve predictive accuracy.

As suggested by Reviewer 1, conduct robustness checks for the Structural Model Assessment [including: Linearity (Quadratic effect), Heterogeneity (FIMIX-PLS), and Endogeneity (Gaussian Copula)].

Response:

Thank you for your valuable suggestions to enhance the methodological rigor of the study. In response:

Model Fit Indices: We have added model fit indices (SRMR, NFI, D_ULS, and D_G) to evaluate the overall fit of the PLS-SEM model, following the guidelines for variance-based SEM. These details have been reported in the revised manuscript (lines 410-412).

R² Value for PsyCap: We acknowledge that the R² value for PsyCap (0.362) indicates a moderate level of explanatory power. This is discussed in the limitation section (lines 520-523), where we explain that the linear assumptions underlying the model may have limited its ability to fully capture complex nonlinear effects, which may partly account for the moderate R² value. We appreciate the suggestion regarding adjusted R²; however, it is not commonly reported in PLS-SEM literature due to the algorithmic differences from OLS-based regression.

Robustness Checks for Structural Model: As suggested by Reviewer 1 and reiterated here, we conducted additional robustness checks to strengthen the validity of the structural model. Specifically, we performed (a) quadratic effect analysis to assess nonlinearity, (b) FIMIX-PLS to explore unobserved heterogeneity, and (c) Gaussian Copula procedures to test for potential endogeneity. These procedures and their results are presented in the revised manuscript (lines 434-470).

4.Response to comment: Conclusion:

Discuss study limitations (e.g., contextual constraints, generalizability issues) more explicitly.

Provide clearer policy recommendations for universities and educational institutions on implementing SDL and PsyCap programs.

 Response:

Thank you for your valuable comments. In response to your suggestions:

Study Limitations: We have revised the manuscript to include a more explicit discussion of study limitations following the discussion section (lines 516-533). These include methodological constraints such as the use of cross-sectional data and reliance on self-report questionnaires, contextual limitations regarding the regional scope of the sample, and the absence of triangulation methods. These limitations are intended to guide future research directions and enhance the transparency and applicability of the findings.

Policy Recommendations: To strengthen the practical contributions of the study, we have expanded the conclusion section to provide clearer and more actionable policy recommendations for universities and educational institutions (lines 549-553 and 562-568). These include suggestions for implementing structured SDL programs, integrating reflective PsyCap development into faculty evaluation systems, and establishing institutional support mechanisms such as resilience workshops and peer coaching groups.

In all, I found the reviewer’s comments are quite helpful, and I revised my paper point-by-point. Thank you and the review again for your help! We tried our best to improve the manuscript and made some changes in the manuscript. We appreciate for Editors/Reviewers’ warm w

---

## [Decision Letter · Decision Letter 1]

Bridging Self-directed Learning and Competency in Interdisciplinary Teaching Among STEM Lecturers: How Psychological Capital Connects

PONE-D-25-02654R1

Dear Dr. Fadzilb,

We’re pleased to inform you that your manuscript has been judged scientifically suitable for publication and will be formally accepted for publication once it meets all outstanding technical requirements.

Kind regards,

Dipendra Karki, Ph.D.

Academic Editor

PLOS ONE

Additional Editor Comments (optional):

Dear Author(s),

Thank you for your thorough revisions, which have significantly improved the clarity and quality of your manuscript. We truly value your careful responses to the reviewers' and editor's comments.

Your manuscript now meets the standards for publication in PLOS ONE.

Decision: Accept

Regards,

Academic Editor

PLOS ONE

Reviewers' comments:

Reviewer's Responses to Questions

**Comments to the Author**

Reviewer #1: All comments have been addressed

Reviewer #2: All comments have been addressed

2. Is the manuscript technically sound, and do the data support the conclusions?

Reviewer #1: Yes

Reviewer #2: Yes

3. Has the statistical analysis been performed appropriately and rigorously?

Reviewer #1: Yes

Reviewer #2: Yes

4. Have the authors made all data underlying the findings in their manuscript fully available?

Reviewer #1: Yes

Reviewer #2: Yes

5. Is the manuscript presented in an intelligible fashion and written in standard English?

Reviewer #1: Yes

Reviewer #2: Yes

Reviewer #1: (No Response)

Reviewer #2: As I thoroughly reviewed the first draft of manuscript and gave my comments and author(s) also rewrote accordingly, now I think it is in a good structure to get published. I urge academic editor and editorial board to have a final check whether or not other required issues of PLOS One have been met in the paper. I accept this paper now.

**Do you want your identity to be public for this peer review?** For information about this choice, including consent withdrawal, please see our Privacy Policy

Reviewer #1: **Yes: ** Assoc.Prof.Dr.Nath Amornpinyo

Reviewer #2: **Yes: ** Devid Kumar Basyal, PHD

---

## [Editor Report · Acceptance letter]

PONE-D-25-02654R1

PLOS ONE

Dear Dr. Fadzil,

I'm pleased to inform you that your manuscript has been deemed suitable for publication in PLOS ONE. Congratulations! Your manuscript is now being handed over to our production team.

Kind regards,

on behalf of

Dr. Dipendra Karki

Academic Editor

PLOS ONE